

# Transcriptomic and metabolomic differences between banana varieties which are resistant or susceptible to Fusarium wilt

Dandan Tian[1], Liuyan Qin[1], Krishan K. Verma[2], Liping Wei[1], Jialin Li[1], Baoshen Li[1], Wei Zhou[1], Zhangfei He[1], Di Wei[1], Sumei Huang[1], Shengfeng Long[1], Quyan Huang[1], Chaosheng Li[1] and Shaolong Wei[3]

[1] Biotechnology Research Institute, Guangxi Academy of Agricultural Sciences, Nanning, China
[2] Sugarcane Research Institute, Guangxi Academy of Agricultural Sciences, Nanning, China
[3] Guangxi Subtropical Crops Research Institute, Naning, China

Corresponding authors
Chaosheng Li, gxxjycglyjs@163.com
Shaolong Wei, nngt51578@126.com

## ABSTRACT

**Background:** Fusarium wilt, caused by *Fusarium oxysporum* f. sp. cubense race 4 (Foc4), is the most lethal disease of bananas in Asia.

**Methods:** To better understand the defense response of banana to Fusarium wilt, the transcriptome and metabolome profiles of the roots from resistant and susceptible bananas inoculated with Foc4 were compared.

**Results:** After Foc4 inoculation, there were 172 and 1,856 differentially expressed genes (DEGs) in the Foc4-susceptible variety (G1) and Foc4-resistant variety (G9), respectively. In addition, a total of 800 DEGs were identified between G1 and G9, which were mainly involved in the oxidation-reduction process, cell wall organization, phenylpropanoid biosynthesis, and lipid and nitrogen metabolism, especially the DEGs of Macma4_08_g22610, Macma4_11_g19760, and Macma4_03_g06480, encoding non-classical arabinogalactan protein; GDSL-like lipase; and peroxidase. In our study, G9 showed a stronger and earlier response to Foc4 than G1. As the results of metabolomics, lipids, phenylpropanoids and polyketides, organic acids, and derivatives played an important function in response to Fusarium wilt. More importantly, Macma4_11_g19760 might be one of the key genes that gave G9 more resistance to Foc4 by a lowered expression and negative regulation of lipid metabolism. This study illustrated the difference between the transcriptomic and metabolomic profiles of resistant and susceptible bananas. These results improved the current understanding of host-pathogen interactions and will contribute to the breeding of resistant banana plants.

## INTRODUCTION

As an important food crop and trade agricultural products around the world, banana is an important cash crop for many countries and plays a key role in economic and social development (*Moffat, 1999*). China is the second-largest banana producer in the world. However, in recent years, banana crops have suffered serious damage from Fusarium wilt

(*Cheng et al., 2017*; *García-Bastidas et al., 2014*). The disease has occurred in almost all banana-producing areas in the world and caused serious harm to the banana industry worldwide.

Fusarium wilt is a soil-borne fungal disease caused by *Fusarium oxysporum* F. Sp. *Cubense* (Foc). According to the host, the pathogen can be divided into physiological races 1 (Foc1), 2 (Foc2), and 4 (Foc4). Foc4 has the widest range of pathogenic hosts and the strongest pathogenic ability (*Cheng et al., 2017*; *Molina, Masdek & Liew, 2001*). The toxin produced by Foc4 plays an important role in the pathogenic process of *F. oxysporum*. It binds to some proteins in the protoplasmic membrane of the cell and thus damages membrane structure and function, leading to changes in membrane permeability, electrolyte leakage, and increased conductance and even the whole plant may wilt (*Sun et al., 2012*). The toxin produced by Foc4 has seven components, the main components are fusarium acid and beauveria (*Xu, Wu & Lin, 2004*; *Li et al., 2010*). The resistant varieties show the strong tolerance to crude toxin and the degree of damage of the ultrastructure of the leaves is relatively mild. The damage inflicted by fusarium acid on the ultrastructure of leaves is like that inflicted by crude toxin, but the degree of damage is mild (*Li et al., 2011*). The proportion of aerenchyma in mature roots of susceptible cultivars is higher than that in resistant cultivars (*Aguilar, Tumer & Sivasithamparam, 2000*).

Whether bananas are disease resistant may also be related to their reaction speed (*Wei et al., 2005*). The amount of reducing sugar released by susceptible banana cultivars after 24 h of pectinase solution treatment was about 1.82 times of that released by resistant banana cultivars (*Li & Zhang, 2010*). The SOD (Superoxide Dismutase) activity of Sanming wild banana root was higher than that of Tianbao banana, indicating that the resistance of Sanming to Fusarium wilt might be related to the higher root system and the up-regulated SOD activity in the early stage (*Cheng et al., 2017*).

After the root of the resistant cultivar was infected with Foc pathogen, the synthesis of phenolic compounds increases rapidly and reach to the peak at about 16 h (*Ana, de Ascensao & Dubery, 2003*). In a banana cultivar inoculated with Foc pathogens, peroxidase increases significantly 6 days after inoculation, while phenylalanine ammonia lyase; chitinase, and β-1, 3-glucanase increased significantly from the third day and reached the peak on the 6[th] day (*Thangavelu et al., 2003*). The activity of exonuclease chitinase is significantly higher in resistant cultivars than that in susceptible cultivars, and there is a positive correlation between the activity of exonuclease chitinase and the resistance and susceptibility of banana cultivars (*Li, Li & Yu, 2010*). Besides, Fusarium wilt infection causes GA3 (Gibberellin A3), IAA (indole-3-acetic acid), and ABA (Abscisic Acid) to accumulate in bananas (*Tang et al., 2006a*).

The root exudates of the susceptible cultivars could significantly promote hyphal growth and spore germination, and the root exudates of both cultivars could significantly promote the growth of and biofilm formation by *B. subtilis* (*Gan et al., 2020*). The root exudates of banana varieties Guijiao 9 and Guijiao 1 were collected. The content of soluble sugar and free amino acids in the root exudates of the resistant cultivar Guijiao 9 has been found to be significantly lower than that of the susceptible cultivars Guijiao 1, but the content of

organic acids is opposite. Phenylalanine and proline have been detected only in the root exudates of the resistant cultivars (*Tian et al., 2017*).

High-throughput sequencing is extensively used to study resistance mechanisms in plants and host-pathogen interaction and identify promising resistant targets (*Niu et al., 2018*). Comparative transcriptomics analysis in the Cavendish banana variety during early infection with Foc1 and Foc4 revealed 1,862 and 226 differentially expressed genes (DEGs), respectively. However, Foc4 infection has almost no effect or a weak effect on these pathways and gene expression (*Dong et al., 2020*). A series of plant-resistance-related proteins are differentially accumulated after infection with both Foc races, indicating that the resistance of the Brazilian variety against two races is different (*Dong et al., 2019*).

Recently, combined analysis of transcriptome and metabolome data has been used to study the environmental adaptation mechanisms of plants to biotic and abiotic stresses (*Zhu et al., 2021*; *Li et al., 2022*; *Hu et al., 2020*). Thus, in this study, we employed joint transcriptomic and metabolomic analyses to identify the resistance mechanism in banana by comparing the difference between the resistant varieties Guijiao 9 (G9) and the susceptible varieties Guijiao 1 (G1) after inoculation with Foc4. This research has a certain guiding significance for studying the influence of Foc4 on banana and the analysis of plant resistance mechanisms, and it provides new insights into the possibilities for improving the tolerance of banana to Fusarium wilt through genetic engineering technology.

# MATERIALS AND METHODS

## Plant material cultivation and Foc4 treatment

In this study, Guijiao 1 (G1, Fusarium wilt susceptible variety) and Guijiao 9 (G9, Fusarium wilt moderately resistant banana variety) were used to performed the transcriptomics and metabolisms experiments. We took the banana seedlings out of the nutrient cups and rinsed them three to five times with tap water to wash the culture substrate, then three to four times with distilled water, and finally three to four times with ultrapure water. We soaked the seedlings in thymol solution (5 mg/mL) for 3 min and then rinsed them with ultrapure water three times.

We placed the banana seedlings in a container that had been sterilized in advance, added 800 mL of Hoagland liquid medium and 50 mL of 5 mmol/L calcium chloride solution, and then added an appropriate amount of ceramist to cover the bare roots. All the plants were placed in a greenhouse and allowed to grow under natural light conditions. After 1 day of culture, the first sample was taken as the control group.

The banana seedlings were removed from Hoagland liquid medium, immersed in 800 mL of Foc4 spore suspension (Hoagland liquid medium) containing spores at $1 \times 10^7 \cdot ml^{-1}$ concentration for 2–2.5 h, and then returned to the original Hoagland liquid medium (containing ceramsite). The second sample was taken on the third day after inoculation with Foc4. The third sample was taken on the seventh days after inoculation.

Root tissues (about 1 cm of the white root tip) were collected and washed quickly with PBS buffer to remove surface impurities of the root, flash-frozen with liquid nitrogen, placed in 1.5 mL tubes precooled in liquid nitrogen, and stored at −80 °C in a refrigerator for RNA extraction, library building, and transcriptome sequencing. The sample was

ground in a homogenizer and then added to Trizol reagent (Invitrogen, Waltham, MA, USA). After vortex mixing, chloroform was added and centrifuged at low temperature. The RNA was precipitated with ethanol and dissolved in Rnase free water. RNA integrity was assessed using the RNA Nano 6000 Assay Kit of the Agilent Bioanalyzer 2100 system (Agilent Technologies, Santa Clara, CA, USA). RNA concentration was measured using Qubit® RNA Assay Kit in Qubit® 2.0 Fluorometer (Life Technologies, Carlsbad, CA, USA). The sample size estimation was estimated by using RnaSeqSampleSize (*Zhao et al., 2018*).

The 20 mL root culture medium was placed in a pre-cooled 50 mL centrifuge tube, flash-frozen in liquid nitrogen, and then stored in a refrigerator at −80 °C. It was used to identify the metabolome of root exudates. After the samples were slowly thawed at 4 °C, appropriate amount of samples were added to pre-cooled methanol/acetonitrile/aqueous solution (2:2: 1, v/v), vortex mixing, ultrasound at low temperature for 30 min, stand at −20 °C for 10 min, centrifuge at 14,000 g and 4 °C for 20 min, vacuum drying with supernatant, adding 100 μL acetonitrile solution during mass spectrometry (acetonitrile: Water = 1:1, v/v) dissolved, swirled, centrifuged 14,000 g at °C for 15 min, and the supernatant was taken for analysis. Transcriptome and metabolome samples were taken at the same time. Three biological replicates were collected for each treatment.

## Sequencing and transcriptome data analysis

The RNA-Seq libraries were prepared using the Illumina TruSeq RNA Sample Pre-Kit following the manufacturer's protocols. The sequencing had been performed in paired-end reads (2 × 151 bp) using the Illumina HiSeq sequencing platform (Beijing Biomics Biotech Co. Ltd., Beijing, China). The RNA-Seq data have been deposited at GenBank under the accession PRJNA916078.

Trimmomatic (version 0.38) was used to remove the adaptor sequences and the low-quality (<Q20) bases at the 5′ and 3′ ends (*Bolger, Marc & Bjoern, 2014*), and reads longer than 70 bp were used for further experiment. The reads were mapped to the banana genome (*D'Hont et al., 2012*) using bowtie2 (version 2.1.0) (*Langmead & Salzberg, 2012*) with default parameters after preprocessing of RNA-Seq data. TopHat2 (version 2.2.1) was also used to perform a sequence comparison between clean reads and the reference genome to obtain position information on the reference genome or gene and sequence characteristic information unique to the sequencing sample. Gene expression levels were presented as FPKM (fragments per kilobase of transcript per million fragments mapped) values calculated by using DESeq2 (version 3.17). Genes with expression levels >1 FPKM were retained for further analysis. We used FDR < 0.05 and more than two-fold change as the criteria to classify DEGs between two compared pairs after the pairs were processed by DEseq2 (version 3.17, www.bioconductor.org). The resulting $p$ values were adjusted using Benjamini and Hochberg's approach for controlling the false discovery rate.

AmiGO (version 2.0) with the default parameters was used to obtain the gene ontology terms of each gene and analyze GO (Gene Ontology) functional enrichment by using hypergeometric tests with FDR (false discovery rate) correction to obtain an adjusted $p < 0.01$ between test gene groups and the whole annotation data set, respectively.

The differentially expressed genes in the KEGG pathway were analyzed using Cytoscape (version 3.10.0) (*Ideker, 2011*) with the ClueGO plugin (*Wegdam et al., 2014*).

## Identification and data analysis of untargeted metabolomics

Samples were separated on an ultra-high-performance liquid chromatography system (UHPLC, Agilent 1290 Infinity LC) HILIC column. The column temperature was 25 °C, the flow rate was 0.5 mL/min, and the injection volume was 2 μl. The mobile phase consisted of two phases: A (water + 25 mM ammonium acetate + 25 mM ammonia water) and B (acetonitrile). The gradient elution procedure was as follows: 0–0.5 min, 95% B; 0.5–7min, B changes linearly from 95% to 65%; 7–8 min, B changes linearly from 65% to 40%; 8–9 min, B maintained at 40%; 9–9.1 min, B changes linearly from 40% to 95%; 9.1–12 min, B maintained at 95%. During the entire analysis, the samples were placed in an autosampler at 4 °C.

To avoid the influence of the fluctuating instrument detection signal, the samples were analyzed continuously in random order. Quality control (QC) samples were inserted into the sample cohort to monitor and evaluate the stability of the system and the reliability of the experimental data.

The AB Triple TOF 6,600 Mass Spectrometer (AB SCIEX) was used to perform mass spectrometry analysis. The ESI Source conditions after HILIC chromatographic separation were as follows: Ion Source Gas1, Gas2 and Curtain Gas (CUR) was of 60, 60 and 30, respectively; source temperature was of 600 °C; IonSapary Voltage Floating (ISVF) was of ± 5,500 V (both positive and negative modes), TOFMS scan M/Z range from 60 to1,000 Da; product ION scan M/Z range from 25 to1,000 Da; TOF MS scan retention time was of 0.20 s/spectra, and product ion scan retention time was 0.05 s/spectra. To acquire high resolution mass spectra, the secondary mass spectra data obtained were obtained using information-dependent acquisition (IDA) of ESI positive ion mode and the ESI negative ion mode. The declustering potential were following as *in situ*: ±60 v. Frontiers: 35 ± 15 eV. The IDA was set as follows: ex situ within 4 Da and candidate ions to monitor per cycle: 10.

The original data was converted into MzXML format by ProteoWizard, and then XCMS software was used for peak alignment, retention time correction and peak area extraction. Firstly, the metabolite structure identification and data preprocessing were carried out for the data extracted by XCMS, and then the experimental data quality evaluation was carried out, and finally the data analysis was carried out. Peak alignment, retention time correction, and peak area extraction were performed by MSDAIL software. Metabolite structure identification, data preprocessing, experimental quality evaluation, and data analysis were carried out for data extracted from MSDAIL. Principal component analysis (PCA) was performed in SIMCA-P 14.1 software to investigate the differences between biological replicates of different samples. Orthogonal partial least squares discriminant analysis (OPLS-DA) was used to measure the differences between the treatment groups. The differential metabolites were screened according to the contribution value (VIP ≥ 1), the significance of the change between groups ($p < 0.05$), and the fold change (fold change ≥ 2).
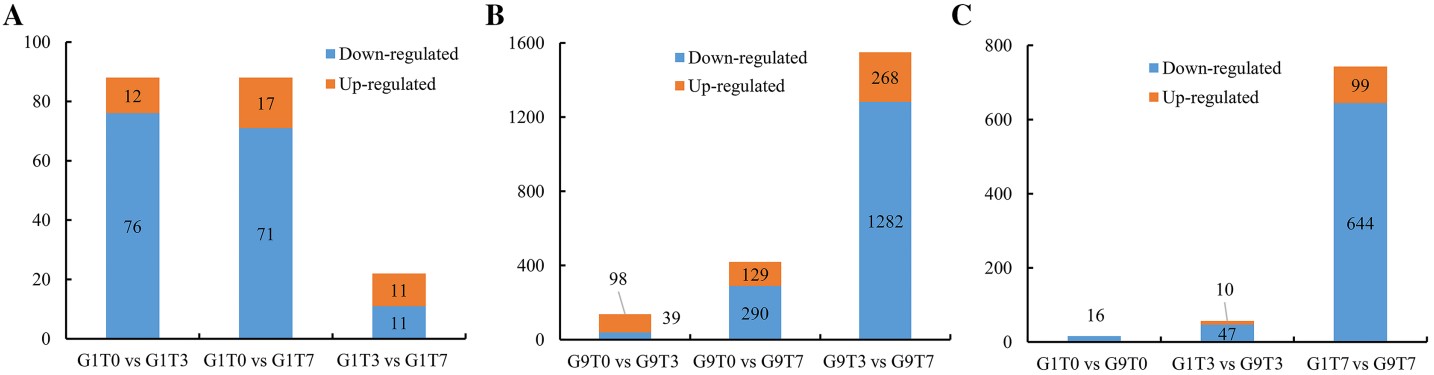

**Figure 1 Statistics of DEGs among comparisons.** (A) The number of DEGs in the G1 variety under Foc4 treatment. (B) The number of DEGs in the G9 variety under Foc4 treatment. (C) A comparison of the number of DEGs in the G1 and G9 varieties under Foc4 treatment.

The data reported in this article have been deposited in the OMIX, China National Center for Bioinformatics/Beijing Institute of Genomics, Chinese Academy of Sciences (https://ngdc.cncb.ac.cn/omix: accession no. OMIX004812).

# RESULTS

To identify the key genes involved in regulating the response to Foc4, we carried out RNA-seq analysis without inoculation and in two stages (3 and 7 days) after Foc4 inoculation of two varieties. After filtering out low-quality reads, we obtained 392,644,830 high-quality clean reads for 18 samples, with an average of 21.81 million clean reads for every sample. More than 90% of these reads successfully mapped to the reference genome (Table S1). We have performed three biological replicates for each treatment sample and the statistical power of this experimental design, calculated in RNASeqPower is 0.96. In addition, we obtained a total of 38,049 expressed genes based on our transcriptome data with the FPKM value from 0 to 9,764.11 (Table S2). All these identified genes were annotated in each database and were presented in Tables S3 and S4. The spearman correlation analysis was used to test the repeatability of experimental results (Fig. S1).

## Identification of Foc4-responsive genes

In the Foc4-susceptible variety (G1), total of 76 genes showed decreased expression and 12 genes showed increased expression after 3 days of Foc4 inoculation, and 71 genes showed decreased expression and 17 genes showed increased expression on seventh day after inoculation. In all, 11 genes were up-regulated and 11 genes were down-regulated on seventh day compared with those on third day after inoculation (Fig. 1A, S2, and Tables S5–S7).

For the Foc4-resistant variety (G9), the expression of 39 genes decreased and 98 genes increased 3 days after inoculation with Foc4 and the expression of 290 genes decreased and 129 genes increased on seventh day after inoculation. However, 1,282 genes were down-regulated and only 268 genes were up-regulated on seventh day compared with those on the 3rd day after inoculation (Figs. 1B, S3, and Tables S8–S10).

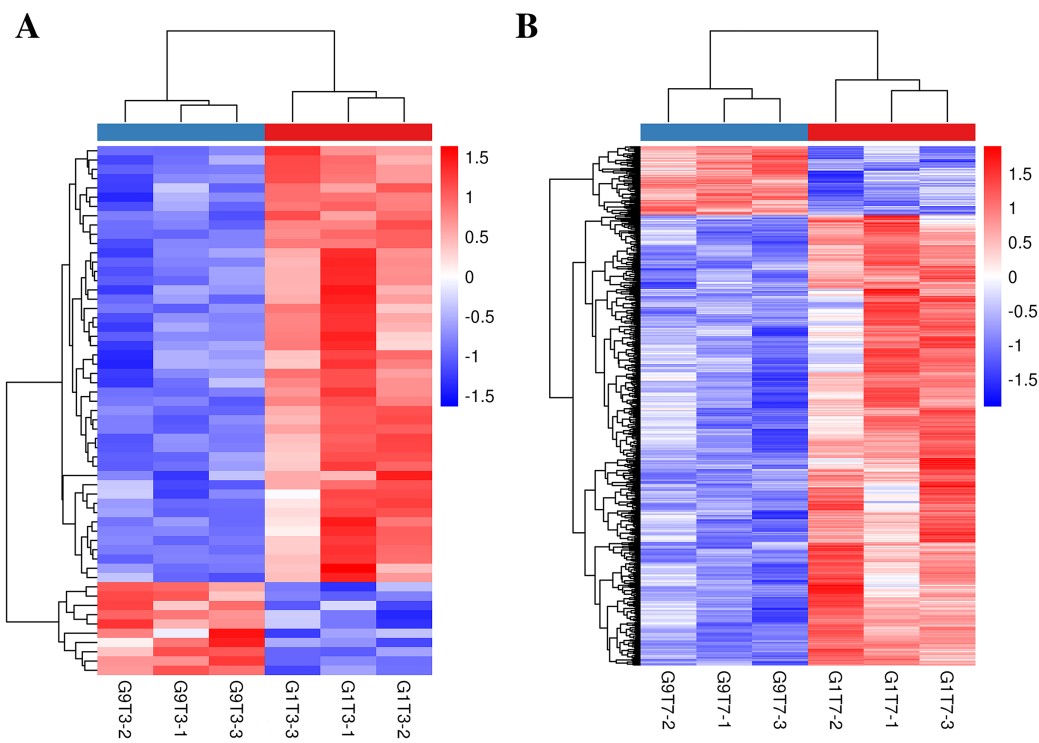

**Figure 2 Heat map of banana genes responding to Foc4 at the transcription level.** (A) A comparison of the DEGs in G1 and G9 varieties under Foc4 treatment for 3 days. (B) A comparison of the DEGs in G1 and G9 varieties under Foc4 treatment for 7 days.

The expression levels of 16 genes in the Foc4-resistant variety (G9) were lower than those in the Foc4-susceptible variety (G1) even before pathogen treatment. On the third day after inoculation, the expression levels of 47 genes in G9 were lower than those in G1 and the expression levels of 10 genes were higher than those in G1 (Figs. 1C, 2A, and Table S11). On the seventh day after inoculation with Foc4, the expression levels of 644 genes in G9 were significantly lower than those in G1 and only the expression levels of 99 genes were higher than those in G1 (Figs. 1C, 2B, and Tables S12 and S13).

## Function annotation of Foc4-responsive genes

To fully analyze the functions of DEGs, we performed GO and KEGG functional enrichment analysis (Figs. 3, 4, S4–S7, and Tables S5–S13). In G1, 3 days after inoculation (Fig. S4A), the DEGs were mainly enriched in the oxidation-reduction process (eight), heme binding (four), the extracellular region (four), the hydrogen peroxide catabolic process (three), and the response to oxidative stress (three) and after 7 days of inoculation (Fig. S4B), the DEGs were mainly enriched in the extracellular region (six), the integral component of the membrane (10), and the cell wall (three) in GO terms.

In G9, 3 days after inoculation (Fig. S5A), the DEGs were mainly enriched in cell wall organization (four), the xyloglucan metabolic process (three), and cell wall biogenesis (three) in GO terms. Compared with the control, 7 days after inoculation, in G9 (Fig. S5B), the DEGs were mainly enriched in the oxidation-reduction process (45), the negative

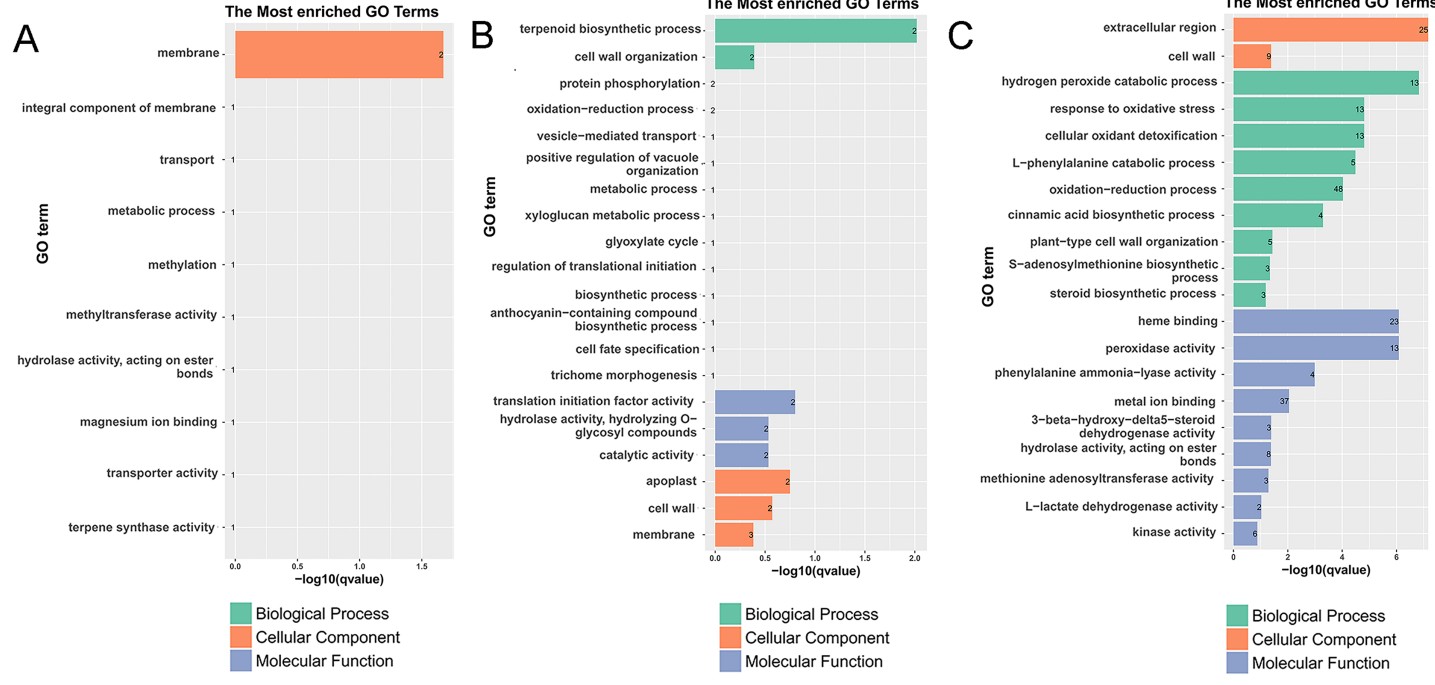

**Figure 3** **Histogram of the most enriched GO classifications of the DEGs.** (A) GO annotation of DEGs between G1 and G9 varieties under normal growth condition. (B) GO annotation of DEGs between G1 and G9 varieties under Foc4 treatment for 3 days. (C) GO annotation of DEGs between G1 and G9 varieties under Foc4 treatment for 7 days.

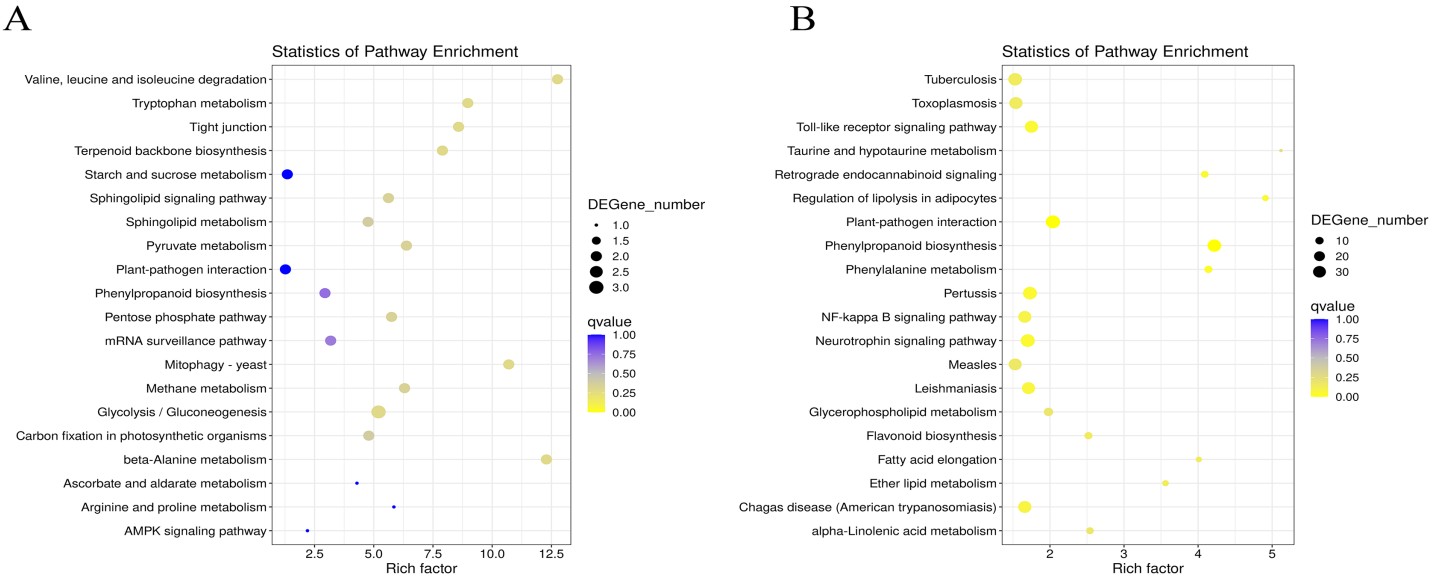

**Figure 4** **Statistics analysis of the most enriched KEGG pathways of the DEGs.** (A) The most enriched pathways of DEGs between G1 and G9 varieties under Foc4 treatment for 3 days. (B) The most enriched pathways of DEGs between G1 and G9 varieties under Foc4 treatment for 7 days.

 

regulation of endopeptidase activity (six), cell wall organization (seven), oxidoreductase activity (14), monooxygenase activity (eight), iron ion binding (12), hydrolase activity (seven), heme binding (12), iron ion binding (12), the extracellular region (14), and the cell wall (seven) in GO terms.

However, compared with three and 7 days after inoculation (Fig. S5C), the DEGs were mainly enriched in the oxidation-reduction process (72), the metabolic process (19), nucleosome assembly (eight), carbohydrate transport (seven), the integral component of the membrane (155), oxidoreductase activity (23), potassium ion transmembrane transporter activity (five), nucleosome (11), and the glucose metabolic process (four). Under normal growth conditions, in G1 and G9, the DEGs were enriched in the membrane (two), methylation (one), the metabolic process (one), and terpene synthase activity (one) (Fig. 3A). In GO terms of 3 days after inoculation, DEGs were enriched in the terpenoid biosynthetic process (two), cell wall organization (two), translation initiation factor activity (two), hydrolase activity (two), the apoplast (two), the membrane (three), the cell wall (two), and catalytic activity (two) (Fig. 3B). After 7 days of inoculation, DEGs were mainly enriched in the extracellular region (25), kinase activity (six), L-lactate dehydrogenase activity (two), methionine adenosyl transferase activity (three), hydrolase activity, acting on ester bonds (eight), metal ion binding (37), phenylalanine ammonia-lyase activity (four), peroxidase activity (13), heme binding (23), the steroid biosynthetic process (three), the S-adenosylmethionine biosynthetic process (three), plant-type cell wall organization (five), the cinnamic acid biosynthetic process (four), the oxidation-reduction process (48), the L-phenylalanine catabolic process (five), cellular oxidant detoxification (13), the response to oxidative stress (13), the hydrogen peroxide catabolic process (13), and the cell wall (nine) (Fig. 3C).

In G1 of 3 days after inoculation (Fig. S6A), the main DEG-enriched KEGG pathways identified were phenylpropanoid biosynthesis; nitrogen metabolism; alanine, aspartate, and glutamate metabolism; and circadian rhythm. After 7 days, phenylpropanoid biosynthesis and plant-pathogen interaction were the mainly enriched KEGG pathways (Fig. S6B).

In G9 of 3 days after inoculation (Fig. S7A), the main DEG-enriched KEGG pathways identified were ubiquitin-mediated proteolysis, the Wnt signaling pathway, plant hormone signal transduction, nitrogen metabolism, and MAPK signaling. After 7 days of inoculation (Fig. S7B), the main DEG-enriched KEGG pathways identified were glycolysis/gluconeogenesis, nitrogen metabolism, starch and sucrose metabolism, and plant hormone signal transduction.

Under normal growth conditions, DEGs between G1 and G9 (Fig. 4A) were enriched in glycerophospholipid metabolism, glycerolipid metabolism and cutin, and suberine and wax biosynthesis KEGG pathways. After 3 days of inoculation, the main DEG-enriched KEGG pathways were glycolysis/gluconeogenesis and terpenoid backbone biosynthesis. After 7 days of inoculation, the main DEG-enriched KEGG pathways were plant-pathogen interaction and phenylpropanoid biosynthesis (Fig. 4B).
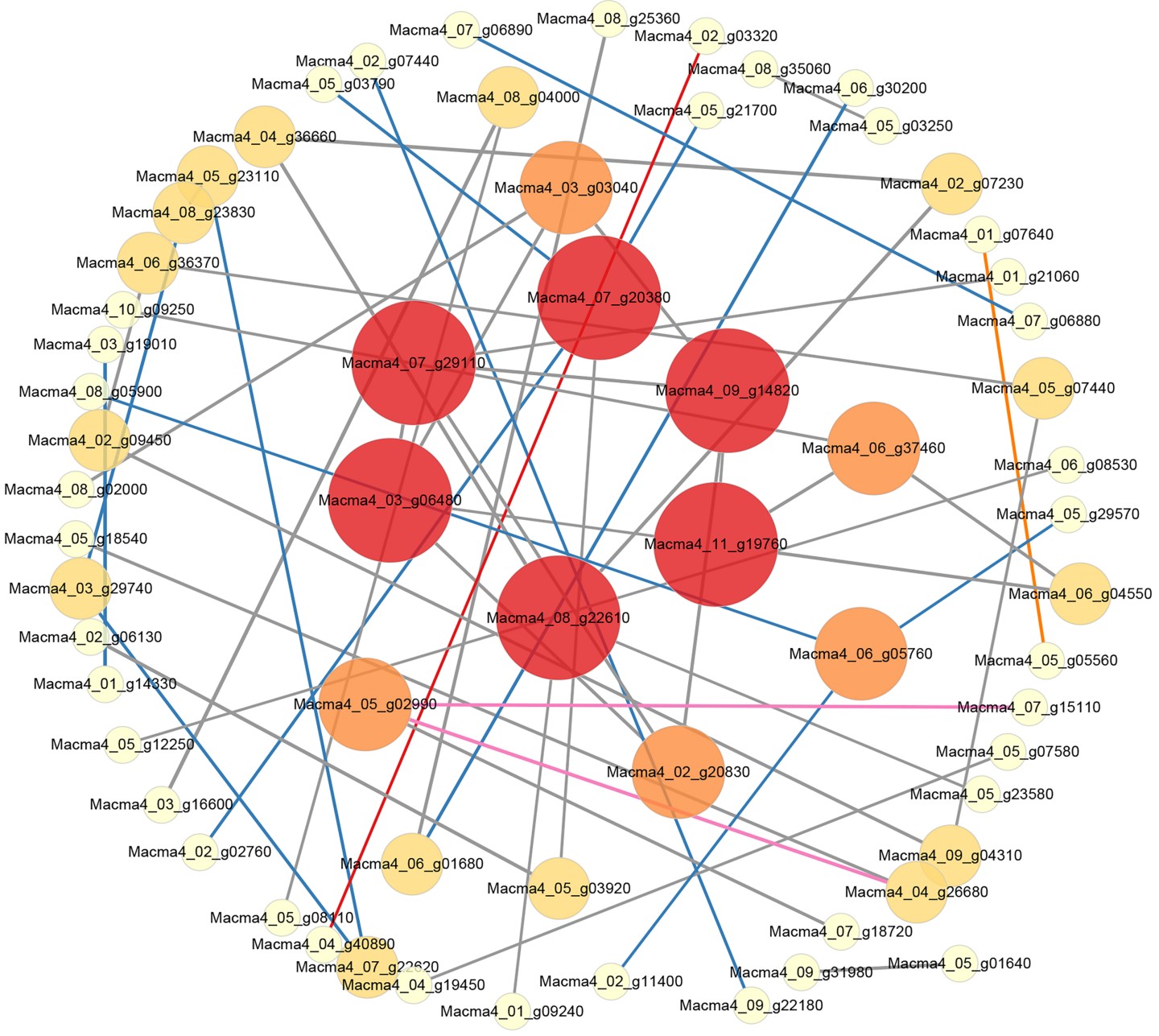

**Figure 5 PPI network of DEGs from G1T7 *vs* G9T7.** The interaction relationship in the STRING protein interaction database (http://string-db.org/) to analyze the differentially expressed genes coding protein interaction network.

## Protein-protein interaction network analysis of Foc4-responsive genes

We constructed an interaction network based on the DEGs retrieved from known PPI databases (Figs. 5, S8 and S9). Regarding G1, when comparing the control and the inoculated G1 variety on third day after inoculation, six DEGs could be linked *via* known nitrogen metabolism relationships. Significantly, the Macma4_09_g15700 (4 link and mainly down-regulated) encoding nitrate reductase was proposed as a potential hub (Fig. S8).

As for G9, when comparing the control and the inoculated variety on third day after inoculation, eight DEGs could be linked *via* known nitrogen metabolism relationships. Macma4_09_g15700 (six link and down-regulated) was also proposed as a potential hub (Fig. S9A). On comparing the control and the inoculated variety 7 days after inoculation (Fig. S9B), 43 DEGs could be linked *via* energy production and conversion, inorganic ion transport and metabolism, and other relationships. Macma4_09_g15510 (nine link and down-regulated) and Macma4_09_g15700 both encoding nitrate reductase were also proposed as potential hubs. In addition, Macma4_10_g32300 (six link and down-regulated) encoding glucose-6-phosphate isomerase, Macma4_07_g06890 (nine link and down-regulated) encoding L-lactate dehydrogenase, and Macma4_09_g25660 pyruvate kinase (nine link and down-regulated) were proposed as the key hub proteins.

When comparing the inoculated varieties on third day after inoculation with those on seventh day after inoculation, 410 DEGs could be linked *via* carbohydrate transport and metabolism, energy production and conversion, and other relationships. Among them, Macma4_11_g21890 (11 link and up-regulated) encoding enolase, Macma4_07_g06890 (10 link and down-regulated), Macma4_02_g23780 (10 link and down-regulated) encoding pyruvate kinase, and Macma4_07_g06880 (10 link and down-regulated) encoding L-lactate dehydrogenase were key potential hubs (Fig. S9C).

Comparing G1 and G9 after inoculation for 7 days, total of 65 DEGs could be linked *via* energy production and conversion, oxidation-reduction reactions, secondary metabolites biosynthesis, and lipid metabolism relationships (Fig. 5).

## Metabolite identification results

In this study, the differences between the metabolites in the two varieties after Foc4 inoculation were analyzed using UPLC-QTOF-MS. The results showed that 400 metabolites were matched with the database in the positive ion mode (Table S14), 237 metabolites were matched in the negative ion mode (Table S15), and 611 metabolites were combined. There are 12 super classes (Table S16) and 82 classes (Table S17).

The peaks extracted from all the experimental samples and QC samples were analyzed by PCA, as shown in Fig. S10. The results showed that QC samples in positive and negative ion modes are closely clustered together, indicating that the experiment has good repeatability. In the OPLS-DA displacement test, the samples of the comparison groups showed an obvious trend of separation, indicated that the model had a good predictive ability and was effective (Fig. S11).

## Differential metabolite screening

According to metabolomics identification, without Foc4 inoculation, there were eight differential metabolites between G9 and G1 in the positive ion mode and two differential metabolites between G9 and G1 in the negative ion mode (Fig. 6). A total of 3 days after Foc4 inoculation, there were four differential metabolites between G9 and G1 in the positive ion mode and five differential metabolites between G9 and G1 in the negative ion mode. On the seventh day after Foc4 inoculation, there were eight differential metabolites
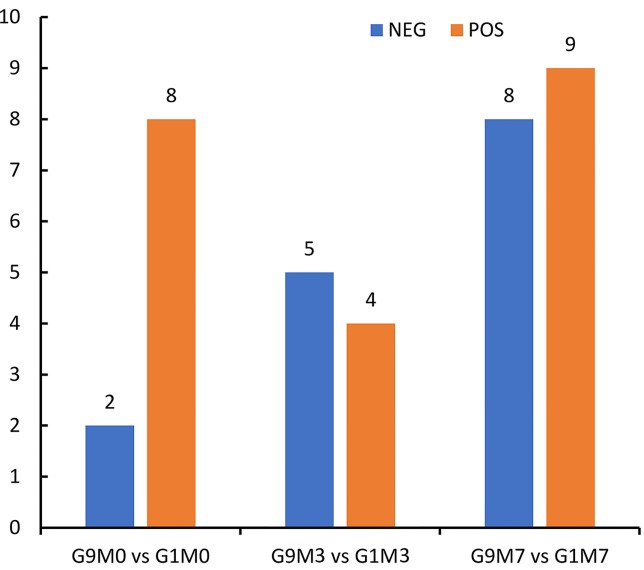

**Figure 6 Statistics of differential metabolites between G1 and G9 under Foc4 inoculation treatment.** NEG and POS refer to the detection under negative and positive ion modes, respectively.

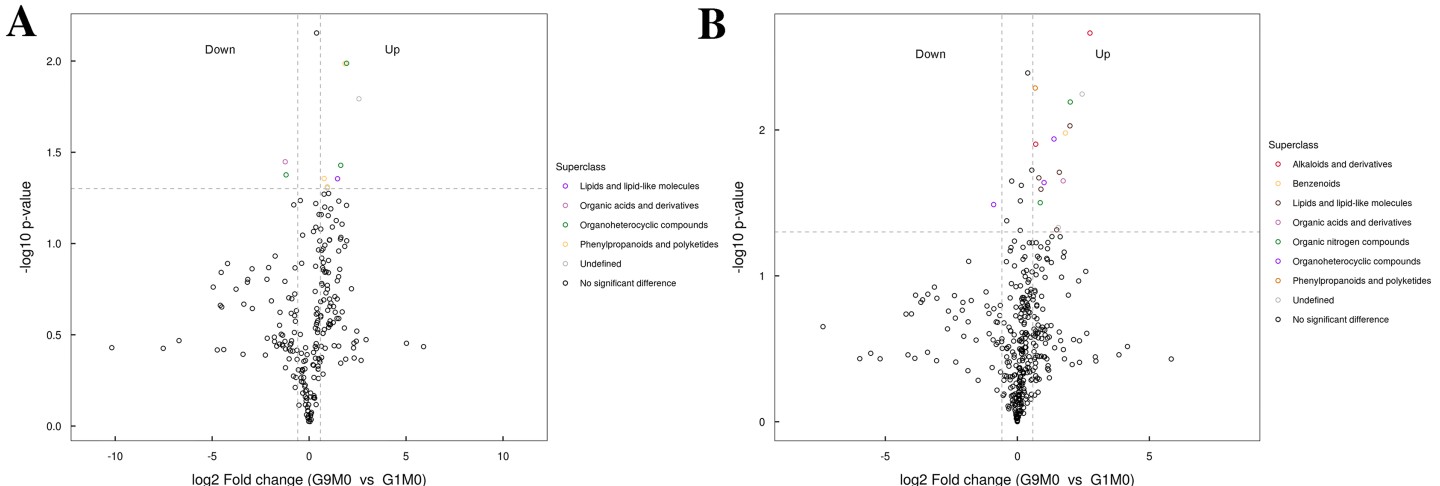

**Figure 7 Volcano plot for the super class statistics of differential metabolites in G9 *vs* that in G1 under normal growth conditions.** (A) Statistics of super class under negative ion modes. (B) Statistics of super class under positive ion modes.

between G9 and G1 in the positive ion mode and nine differential metabolites between G9 and G1 in the negative ion mode.

All the metabolites detected in positive and negative ion modes (including those not identified) were analyzed for differences based on univariate analysis. When the difference between groups was FC > 1.5 or FC < 0.67 and the *p* value < 0.05, the metabolites were defined as differential metabolites and super class classification statistics were performed for the differential metabolites (Figs. 7–9).

In G9 and G1, without Foc4 inoculation, the up-regulated metabolites identified in the negative ion mode mainly belonged to lipids and lipid-like molecules, phenylpropanoids
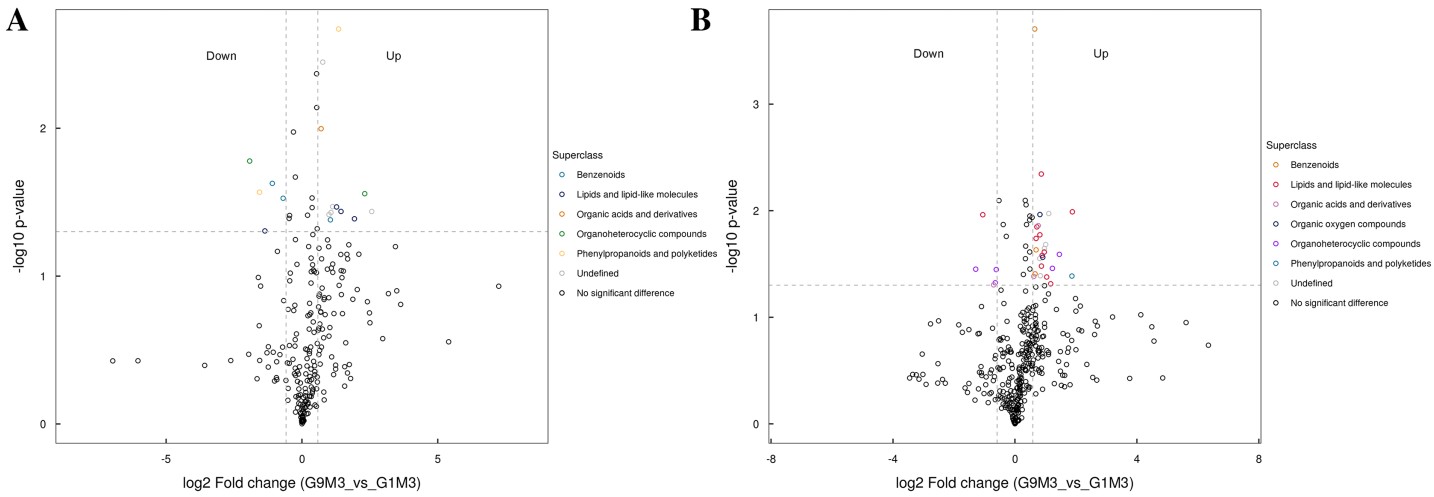

**Figure 8 Volcano plot for the super class statistics of differential metabolites in G9 *vs* that in G1 under Foc4 treatment for 3 days.** (A) Statistics of super class under negative ion modes. (B) Statistics of super class under positive ion modes.

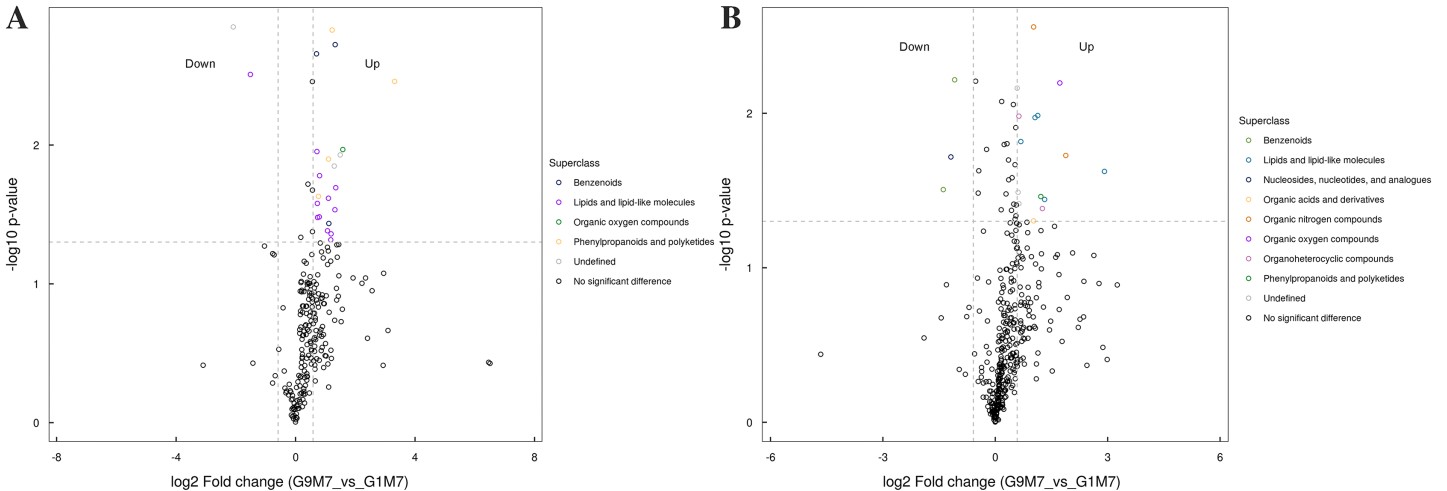

**Figure 9 Volcano plot for the super class statistics of differential metabolites in G9 *vs* that in G1 under Foc4 treatment for 7 days.** (A) Statistics of super class under negative ion modes. (B) Statistics of super class under positive ion modes.

and polyketides, and organic acids and derivatives (Fig. 7). The metabolites identified in the positive ion mode mainly belonged to alkaloids and derivatives, benzenoids, lipids and lipid-like molecules, organic acids and derivatives, organic nitrogen compounds, and phenylpropanoids and polyketides. The up-regulated metabolites were mainly of the organ heterocyclic compounds.

After 3 days of Foc4 inoculation, the differential metabolites identified in positive and negative ion modes were similar in classification and the up-regulated metabolites mainly belonged to benzenoids, lipids and lipid-like molecules, organic acids and derivatives, the organ heterocyclic compounds, and phenylpropanoids and polyketides (Fig. 8).

After 7 days of Foc4 inoculation, the up-regulated metabolites identified in the negative ion mode were mainly benzenoids, lipids and lipid-like molecules, organic oxygen

compounds, phenylpropanoids and polyketides (Fig. 9). The metabolites identified in the positive ion mode mainly belonged to lipids and lipid-like molecules, organic oxygen compounds, and phenylpropanoids and polyketides. The up-regulated metabolites mainly belonged to benzenoids and nucleosides, nucleotides, and analogues.

## DISCUSSION

Banana Fusarium wilt is the most widely distributed banana disease and the biggest threat to the banana industry (*García-Bastidas et al., 2014*). Disease resistance is the result of co-evolution, mutual selection, and mutual adaptation in the long-term interaction between plants and their pathogenic organisms (*Li, Shao & Wang, 2013*; *Li et al., 2012*). After the pathogen invades the plants, it produces substances such as hormones, enzymes, and toxins, which harm the plant. Protectases and phenolic compounds was able to prevent the spread of Foc4 and achieve disease resistance (*Kuc & Rush, 1985*; *Caracuel et al., 2003*). The increase in ferulic acid content was able to increase the lignification degree of cell wall, playing an important role in plant disease resistance (*Asunción García-Sánchez et al., 2010*; *Denisov, Freeman & Yarden, 2011*). After Foc4 infection, the gene expression of phenylamine lyase, cinnamyl alcohol dehydrogenase, peroxidase, polyphenol oxidase, and other enzymes involved in cell wall reinforcement are also up-regulated to varying degrees (*De Ascensao & Dubery, 2000*). High levels of phenolic substances were related to Fusarium wilt infection in banana (*Tang et al., 2006a*; *Tang et al., 2006b*). RNA-seq results indicates that basal defense mechanisms are involved in pathogen-associated molecular pattern (PAMP) recognition, activation of effector-triggered immunity (ETI), ion influx, and biosynthesis of hormones as well as pathogenesis-related (PR) genes (*Dong et al., 2020*).

In our study, two varieties showed great difference to Foc4 (Fig. S12). In this study, when susceptible variety G1 and resistant variety G9 were infected by Fusarium wilt, the DEGs in both were annotated with oxidation-reduction process, cell wall organization, the catabolic process, and oxidoreductase activity (Figs. S4 and S5). The main DEG-enriched KEGG pathways were phenylpropanoid biosynthesis, plant hormone signal transduction, and nitrogen metabolism (Figs. S6 and S7). These results were consistent with those of previous reports.

When the two varieties were compared, we found that 3 days after inoculation with Foc4, there was an increase in the number of expressed genes (98) in the resistant variety of G9, which was significantly more than the number of expressed genes (12) in the susceptible cultivars G1 (Figs. 1A and 1B), suggesting that the reaction of G9 against with the pathogen was faster, although they involved the similar pathways and there was little difference in terms of DEGs. However, after 7 days of pathogen inoculation, 268 genes were up-regulated and as many as 1,282 genes were repressed in G9. In terms of the number of DEGs, G9 had far more than that of G1, indicating that G9 had a stronger response to Foc4.

According to the results of GO annotation, there was a difference between G1 and G9 under normal growth conditions, which referred to membrane enrichment, methylation, the metabolic process, and terpene synthase activity (Fig. 3). After 3 days of inoculation,

the difference was concentrated in the terpenoid biosynthetic process, cell wall organization, translation initiation factor activity, hydrolase activity, the apoplast, the membrane, and the cell wall. After 7 days of inoculation, we found that DEGs were mainly enriched in the extracellular region, methionine adenosyl transferase activity, metal ion binding, phenylalanine ammonia-lyase activity, the steroid biosynthetic process, the cell wall, the cinnamic acid biosynthetic process, the oxidation-reduction process, cellular oxidant detoxification, the response to oxidative stress, and the hydrogen peroxide catabolic process. According to the results of KEGG annotation, the main difference between G1 and G9 was enriched glycolysis/gluconeogenesis, terpenoid backbone biosynthesis, plant-pathogen interaction, and phenylpropanoid biosynthesis (Fig. 4).

According to protein interaction network analysis, inoculation of G1 by Foc4 mainly affected the nitrogen metabolism pathway. In addition to nitrogen metabolism, energy production and conversion, transport and metabolism and other relationships in G9, as well as more additional hub proteins were identified (Figs. S8 and S9). On comparing G1 and G9 after inoculation of 7 days, DEGs were found to be linked *via* energy production and conversion, oxidation–reduction reactions, biosynthesis of secondary metabolites, and lipid metabolism relationships (Fig. 5). In particular, for Macma4_08_g22610 (encoding non-classical arabinogalactan protein 30-like), Macma4_03_g06480 (peroxidase), Macma4_09_g14820 (peroxidase), Macma4_07_g29110 (nucleobase-ascorbate transporter LPE1), Macma4_07_g20380 (trans-cinnamate 4-monooxygenase), and Macma4_11_g19760 (GDSL-like lipase) key potential hubs, all of them had lower expression in G9.

Metabolomics results also confirmed these results of DEGs. Based on the results of metabolomics, both under normal growth conditions and after Foc4 inoculation, all the differential metabolites between G1 and G9 belonged to lipids and lipid-like molecules, phenylpropanoids and polyketides, organic acids and derivatives, alkaloids and derivatives, and benzenoids (Figs. 7–9). Previous reports also confirm our findings. Overexpression of OsGLIP1 and OsGLIP2 reduced resistance to bacterial and fungal pathogens in rice, indicating that these two lipases function as negative regulators of disease resistance (*Gao et al., 2017*). GDSL can also regulate rice root growth by regulating the ethylene signal in rice roots (*Zhao et al., 2020*). All these results have proved again that lipids and their derivatives are the important regulatory factors in the plant defense system, cutin and wax are physical barriers to pathogen infection, and GDSL lipase (encoding by Macma4_11_g19760) could regulate the immune activity in rice through lipid homeostasis.

## CONCLUSIONS

In this study, we combined transcriptomics and metabolomics to better understand the differences between resistant and susceptible bananas against Foc4. Our results showed that when G1 and G9 were infected with Foc4, the DEGs participating in lipid metabolism, phenylpropanoid metabolism, and cell wall modification in the resistant variety G9 displayed an earlier response and were significantly up-regulated compared to the DEGs in the susceptible variety G1. The DEGs of Macma4_08_g22610, Macma4_11_g19760, and

Macma4_03_g06480 were proposed as the key candidate genes in G9 with more resistance to Foc4. The results of this study provided new clues for understanding the mechanism of the resistance of the banana to Foc4 and will offer important principle guidelines for the study of the interaction mechanism between Foc4 and bananas, the cultivation of resistant varieties and the disease control of banana Fusarium wilt.

## ACKNOWLEDGEMENTS

We are grateful to the Guangxi Academy of Agricultural Sciences, Nanning, Guangxi, China, for providing the necessary facilities.

### Funding

This research was funded by the Natural Science Foundation of Guangxi (2021GXNSFAA196050), the Guangxi Innovation Driven Development Program (GuikeAA20302016-4), the Guangxi Key Research & Development Program (GuikeAB19245026), the Guangxi Innovation Driven Development Program (Guike20108005-2), and the National Key Research & Development Program (2020YFD1000104-04). The funders had no role in study design, data collection and analysis, decision to publish, or preparation of the manuscript.

### Grant Disclosures

The following grant information was disclosed by the authors:
Natural Science Foundation of Guangxi: 2021GXNSFAA196050.
Guangxi Innovation Driven Development Program: GuikeAA20302016-4.
Guangxi Key Research & Development Program: GuikeAB19245026.
Guangxi Innovation Driven Development Program: Guike20108005-2.
National Key Research & Development Program: 2020YFD1000104-04.

### Competing Interests

The authors declare that they have no competing interests.

### Author Contributions

- Dandan Tian conceived and designed the experiments, performed the experiments, analyzed the data, prepared figures and/or tables, authored or reviewed drafts of the article, and approved the final draft.
- Liuyan Qin performed the experiments, analyzed the data, prepared figures and/or tables, authored or reviewed drafts of the article, and approved the final draft.
- Krishan K. Verma performed the experiments, prepared figures and/or tables, authored or reviewed drafts of the article, and approved the final draft.
- Liping Wei performed the experiments, analyzed the data, prepared figures and/or tables, and approved the final draft.
- Jialin Li performed the experiments, analyzed the data, prepared figures and/or tables, and approved the final draft.

- Baoshen Li analyzed the data, prepared figures and/or tables, and approved the final draft.
- Wei Zhou performed the experiments, analyzed the data, authored or reviewed drafts of the article, and approved the final draft.
- Zhangfei He performed the experiments, analyzed the data, prepared figures and/or tables, and approved the final draft.
- Di Wei analyzed the data, prepared figures and/or tables, and approved the final draft.
- Sumei Huang conceived and designed the experiments, prepared figures and/or tables, authored or reviewed drafts of the article, and approved the final draft.
- Shengfeng Long performed the experiments, prepared figures and/or tables, authored or reviewed drafts of the article, and approved the final draft.
- Quyan Huang performed the experiments, authored or reviewed drafts of the article, and approved the final draft.
- Chaosheng Li conceived and designed the experiments, performed the experiments, analyzed the data, prepared figures and/or tables, authored or reviewed drafts of the article, and approved the final draft.
- Shaolong Wei conceived and designed the experiments, performed the experiments, analyzed the data, authored or reviewed drafts of the article, and approved the final draft.

## DNA Deposition
The following information was supplied regarding the deposition of DNA sequences:

The datasets GENERATED for this study can be found in the PRJNA916078

## Data Availability
The sequence data is available at NCBI BioProject PRJNA916078; SRR22924224, SRR22924223, SRR22924214, SRR22924213, SRR22924212, SRR22924211, SRR22924210, SRR22924209, SRR22924208, SRR22924207, SRR22924222, SRR22924221, SRR22924220, SRR22924219, SRR22924218, SRR22924217, SRR22924216, SRR22924215.

## Supplemental Information
Supplemental information for this article can be found online at http://dx.doi.org/10.7717/peerj.16549#supplemental-information.

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
