# Peer review of "Transcriptomic and metabolomic differences between banana varieties which are resistant or susceptible to Fusarium wilt"

_PeerJ, doi:10.7717/peerj.16549_

## Round 0.1 · original submission · Major Revisions

The authors are requested to incorporate the reviewers' comments.

Reviewer 1 ·

Basic reporting

This manuscript is well-structured, and the language is clear and concise. The authors provide a clear background of the Fusarium wilt disease, its impact on bananas, and the current state of knowledge regarding disease resistance. I have two minor suggestions for potential improvement:
• The authors may need to double-check all the abbreviations and make sure each of them is clearly defined and/or referenced at its first occurrence. For instance, SOD (Superoxide Dismutase) in Line 64 was not defined; KEGG (Kyoto Encyclopedia of Genes and Genomes) in Line 149 was neither defined nor referenced. GO (Gene Ontology) in Line 150: Even though it is a standard term in genomics, it's good practice to define it when it's first used.
• Line 192, 196, 197…, the authors should use the term “Table S1”, instead of “Supplementary Table 1”, to avoid confusion.

Experimental design

The experimental design appears to be robust, appropriate for the research question and is most clearly described. The use of transcriptomics and metabolomics to analyze the differences between the resistant and susceptible banana varieties is commendable. However, I do have some major concerns and minor suggestions as follows.
• Major Concerns:
a) The authors state in Line 157 that a "C-18" column was used for the procedure, however, later in Line 167, they mention a "HILIC chromatographic separation". This ambiguity is significant because the chemistry of the column can greatly influence the metabolomic profiling results. The C-18 column, typically used in reversed-phase chromatography, relies on the hydrophobic interaction between the column stationary-phase and analytes to achieve the separation of analytes. On the other hand, HILIC (Hydrophilic Interaction Liquid Chromatography) is characterized by a polar stationary phase and is generally used for the separation of polar metabolites. The apparent contradiction between these two different types of columns can impact the interpretation of metabolomics profiling results. It is critical for the authors to specify which column was used at each stage of the analysis and explain their choice. Were different columns used for different classes of metabolites? If so, it should be explicitly stated and justified. The authors are thus strongly encouraged to clarify this point in their methodology to ensure the accuracy of their experimental procedures and the validity of their conclusions.
b) Untargeted metabolomics inherently generates a large volume of data with potential redundancies and can lead to challenges in interpreting the results. Furthermore, many metabolites do not readily produce useful gas-phase fragments that could be used as MS2 fingerprints, which complicates the identification process. The use of a 4Da quadrupole isolation window in IDA scans (Line 174), as stated in your methodology, might also lead to the co-elution of precursor ions. This could result in chimeric MS2 spectra, further complicating the task of metabolite identification. It is therefore important that the authors address how they handled these potential issues in their analysis.
c) Moreover, the manuscript would greatly benefit from a more detailed account of the data processing parameters used in the MS-DIAL analysis (Line 179). This will allow for a better understanding of how the authors addressed these potential pitfalls and will also add to the reproducibility of the study by other researchers. Providing data processing/spectra filtering details such as the retention time alignment tolerance, mass accuracy tolerance, minimum peak intensity, and signal-to-noise ratio would strengthen the manuscript. Please provide this additional information so that the readers can fully understand and replicate the approach if desired.
• Minor suggestions:
a) A more detailed description of the sampling methodology would be beneficial. More technical details on RNA and metabolite extraction (Line 117 to 124) and information/discussion on the number of technical replicates (in contrast to biological replicates), such as the number of LC-MS injections per sample, and the impact on method reproducibility and variability.
b) The reasoning behind the choice of banana varieties G1 and G9 should be more explicit. Why these varieties? Are they representative of a broader range of banana varieties?

Validity of the findings

The findings of the study appear valid and in line with the data presented. The statistical analysis is rigorous and supports the conclusions drawn. However, as is common with such studies, there are potential limitations that should be noted:
• The study findings are based on two banana varieties. Expanding this study to include a larger variety of bananas could increase the robustness and general applicability of the findings.
• While the study identifies certain genes and metabolic pathways involved in resistance, the specific mechanisms by which these genes confer resistance remains unclear. Further studies would be needed to fully understand these mechanisms and their potential for use in breeding resistant banana varieties.

Additional comments

This study makes a substantial contribution to understanding the complex biological interactions between bananas and Fusarium wilt. However, I recommend the following:
• The discussion could benefit from more detailed consideration of how these findings fit into the larger context of plant disease resistance, and what implications they may have for strategies aimed at improving disease resistance in other crops.
• More elaboration on how these findings might translate into practical applications in banana cultivation and breeding would strengthen the paper's impact.
Overall, this manuscript is a valuable contribution to the field and presents intriguing findings. I recommend that it be accepted after minor revisions. Specifically, the authors should address the need for more detailed methodological descriptions and discussion of the broader implications of the findings. The authors should also acknowledge potential caveats such as limited variety sampling and the need for further research to fully understand the mechanisms of resistance.

Reviewer 2 ·

Basic reporting

In the manuscript entitled “Transcriptomic and metabolomic analyses reveals the distinct resistance to Fusarium Wilt between resistant and susceptible banana varieties”, the authors performed comparative RNAseq analysis and metabolic profiling on Fusarium (Foc4) resistant and susceptible banana varieties, to investigate the mechanism of resistance against Fusarium.
1. Overall, the manuscript needs thorough copy-editing before the next submission. There are too many typographical and grammatical errors in this manuscript, and the reviewer cannot list them one-by-one.
2. To demonstrate the infection progress of Foc4, and the decease resistance of the banana varieties used in this study, the authors should include phenotypic pictures of the resistant and susceptible banana seedlings at different time points before and after infection. The Figure S12 only displayed pictures of banana plants 35 days after inoculation, which is not the time point used for sample collection for RNAseq analysis and metabolic profiling.
3. The Introduction needs to be rewritten to have a clear logical flow. And the authors need to summarize the previous studies in more details.
4. Figure 3 and 4 need to be remade. The fonts are too small that these figures are unreadable.
5. For Figure S1, the spearman correlation analysis plot, the authors need to reorder the samples in a sequential order of the replicates at different time points. Also, what is G1T12 and G9T12 in this plot? These samples were not mentioned elsewhere in this manuscript.
6. In the title, “wilt” should not be italicized nor capitalized.

Experimental design

1. In the “Sequencing and Transcriptome data analysis”, line 135–144, bowtie2 and TopHat2 are both read aligners and perform same functions. Which one of the packages was actually used? In addition, the authors need to specify which package was used to calculate the FPKM values.
2. For all the packages used in the RNAseq and bioinformatics analysis, the authors need to provide correct citations of the original publication reporting these packages.

Validity of the findings

No comment.

·

Basic reporting

Manuscript can be accepted

Experimental design

Acceptable

Validity of the findings

NA

Additional comments

NA

---

## Round 0.2 · Minor Revisions

Authors are requested to revise the manuscript as per the suggestions of the reviewers.

Reviewer 1 ·

Basic reporting

I appreciate authors’ efforts to revise the description HILIC method (Line 168-172) in the manuscript. Nonetheless, upon careful evaluation, there are key aspects that should be addressed to ensure this study’s adherence to standard HILIC practices.
HILIC Separation Mechanism:
1. Polar Stationary Phase: In HILIC, the stationary phase is inherently polar, commonly consisting of silica particles or other functional groups. When subjected to a high organic content, this polar stationary phase attracts a distinct water-rich layer.
2. Partitioning Principle: Analytes primarily partition between the organic mobile phase and this water-enriched layer on the stationary phase. Polar compounds tend to interact and reside longer within this water layer, leading to their retention.
3. Water Layer Dynamics: The thickness and properties of this water layer can be influenced significantly by the choice of organic solvent. For instance, acetonitrile tends to promote a more defined water-rich layer compared to methanol, influencing retention times and resolution of analytes.
Given the core mechanism:
1. Gradient Protocol in HILIC: Your described gradient starts with 5% methanol, suggesting a high aqueous phase at onset. Standard practice in HILIC dictates beginning with a high organic content and transitioning to a higher aqueous percentage. This conventional gradient is rooted in the mechanism, ensuring polar analytes are efficiently retained initially, followed by a systematic elution.
2. HILIC Solvent Strength Hierarchy: The solvent strength in HILIC is pivotal: water > methanol > acetonitrile. Your choice of solvents can drastically affect the separation mechanism and the resulting chromatograms.
3. Optimal Mobile Phase Choices: The best practice in HILIC entails:
- Strong solvent: An aqueous buffer (often infused with salts or modifiers for pH adjustments or enhanced ionization).
- Weak solvent: Acetonitrile, due to its ability to form a robust water layer on the stationary phase, enhancing the partitioning mechanism.
4. Methanol vs. Acetonitrile: Choosing methanol over acetonitrile requires careful consideration. Methanol, due to its higher polarity, acts as a stronger eluent in HILIC, potentially reducing retention of polar analytes. Acetonitrile's attributes, especially its ability to foster a distinct water layer on the stationary phase, make it a staple in HILIC, ensuring improved retention and peak resolution.
In light of these insights, I urge you to re-examine both your gradient protocol and solvent choices. By aligning with established conventions in HILIC, you may ensure the credibility and impact of your study.

Experimental design

No more comments

Validity of the findings

No more comments

Additional comments

No more comments

Reviewer 2 ·

Basic reporting

The authors addressed the concerns of this reviewer. However, the reviewer forgot to mention the issues of data deposition in the original review. The RNA-seq reads and metabolic data in this study were not deposited, thus not in compliance with the data deposition requirements of the journal. The authors need to deposit the data into publicly accessible databases and provide accession code in the manuscript.

Experimental design

No comment.

Validity of the findings

No comment.

---

## Round 0.3 · Minor Revisions

The authors are requested to address the comments of Dr. Robert Winkler, the Section Editor:............................................
> The paper is technically sound, but the title should be reconsidered. E.g. "Transcriptomic and metabolomic differences between banana varieties which are resistant or susceptible to Fusarium wilt".

> The metabolomics part needs to be revised. Why "secondary mass spectrometry"? What were the acceptance criteria for identifying metabolites (mass tolerances, etc.).? In this part, there are also many typos, e.g. Mz XML is mzXML; ProteoWizard, etc.

---

## Round 0.4 · accepted · Accept

The manuscript is recommended for acceptance in its current form.